# Relationship between Determinants of Health, Equity, and Dimensions of Health Literacy in Patients with Cardiovascular Disease

**DOI:** 10.3390/ijerph17062082

**Published:** 2020-03-20

**Authors:** Ana Cristina Cabellos-García, Enrique Castro-Sánchez, Antonio Martínez-Sabater, Miguel Ángel Díaz-Herrera, Ana Ocaña-Ortiz, Raúl Juárez-Vela, Vicente Gea-Caballero

**Affiliations:** 1Hospital Universitari I Politècnic La Fe, Health Research Institut La Fe, Research Group GREIACC, 46026 Valencia, Spain; anacris095@gmail.com; 2National Institute for Health Research, Health Protection Research Unit in Healthcare Associated Infections and Antimicrobial Resistance at Imperial College, London SW7 2AZ, UK; 3Nursing Department, Universitat de València, Hospital Clínico Universitario, 46010 València, Spain; 4Equipo de Atención Primaria Sant Ildefons-Cornella 2, Institut Català de la Salut, 08004 Barcelona, Spain; madiazher@gmail.com; 5Hospital Universitari General de Catalunya, Grupo Quirónsalud, 08195, Barcelona, Spain; 6Fundación para el Fomento de la Investigación Sanitaria y Biomédica de la Comunitat Valenciana FISABIO, 46035 València, Spain; a.ocanao@gmail.com; 7International Univeristy of La Roja, Instituto de Investigación Sanitaria de Aragón, 26006 Logroño, Spain; raul.juarez@unirioja.es; 8Health Research Institute of Aragon, 50009 Zaragoza, Spain; 9Escuela de Enfermería La Fe, Adscript Center of Universitat de Valencia, Health Research Institut La Fe, Research Group GREIACC, 46026 Valencia, Spain; gea_vic@gva.es

**Keywords:** health literacy, cardiovascular diseases, social determinants of health, health status disparities, equity

## Abstract

**Background**: Health literacy (HL) has been linked to empowerment, use of health services, and equity. Evaluating HL in people with cardiovascular health problems would facilitate the development of suitable health strategies care and reduce inequity. Aim: To investigate the relationship between different dimensions that make up HL and social determinants in patients with cardiovascular disease. **Methods**: Observational, descriptive, cross-sectional study in patients with cardiovascular disease, aged 50–85 years, accessing primary care services in Valencia (Spain) in 2018–2019. The Health Literacy Questionnaire was used. **Results**: 252 patients. Age was significantly related with the ability to participate with healthcare providers (*p* = 0.043), ability to find information (*p* = 0.022), and understanding information correctly to know what to do (*p* = 0.046). Level of education was significant for all HL dimensions. Patients without studies scored lower in all dimensions. The low- versus middle-class social relationship showed significant results in all dimensions. **Conclusions**: In patients with cardiovascular disease, level of education and social class were social determinants associated with HL scores. Whilst interventions at individual level might address some HL deficits, inequities in access to cardiovascular care and health outcomes would remain unjustly balanced unless structural determinants of HL are taken into account.

## 1. Background

The social determinants of health are defined as those circumstances in which people are born, grow up, live, work, and grow old [1]. The unequal distribution of social determinants results in unfair, systematic, and avoidable health inequalities between socially, economically, demographically, or geographically defined population groups. Inequalities in health are the result of the unequal distribution of power and resources according to gender, social class, territory, or ethnicity, producing a worse state of health in the less socially advantaged citizens [2].

Various models exist to explain the determinants of health inequalities. In Spain, the Commission for Reducing Health Inequalities developed a model based on Navarro and Solar and Irwin for the World Health Organization (WHO) Commission on Social Determinants (Figure 1) [3,4].

This model is composed of structural and intermediate determinants. The former include the political and socio-economic context and the social structure that includes axes of inequality as generators of social power hierarchies, such as gender, age, social class, ethnicity, and territory or environment.

The intermediate determinants are, on the other hand, the material circumstances such as housing and working conditions that affect the psychosocial processes and behaviours of individuals and communities, including the health system and its actors [4]. Structural factors associated with socio-economic components, environment, and the health system are the basis of health inequalities. Socio-economic differences are related to life expectancy, all causes of mortality, and self-perceived health [3]. Inequalities between men and women, between more and less favoured social classes, and educational level have a negative impact on the health of the population, the distribution of the disease, and its risk factors. They also condition life habits [2].

A strong indicator of individual health status, built on social location factors and on interaction with the health system factors, is health literacy (HL) [5]. HL implies the knowledge, motivation, and skills to access, understand, comprehend, and apply health information for decision making related to health care, disease prevention, and health promotion to maintain or improve the quality of life over the life course. The results of the European Health Literacy Survey confirm that HL could be considered as a determinant of people’s health since it does not depend only on one’s own abilities but on the interaction between these individual abilities, social relations, and the health system [5]. It is therefore an effective tool for empowering people and enabling them to exercise greater control over health and its social determinants [6,7,8].

The concept of HL encompasses different dimensions, such as the aforementioned competencies to understand, evaluate, and use health information for decision making; the use of preventive and health services; the environment; social support; and individual and collective empowerment [9,10,11,12]. Increasing evidence demonstrates the impact that HL has on patient behaviour; people with lower levels of HL have greater difficulties in planning and adjusting their lifestyle in the context of chronic illness, in making decisions, and in knowing when to access health services. It is also an important factor in the prevention of chronic non-communicable diseases in terms of its relationship with behavioural determinants, such as lack of physical activity, unhealthy dietary habits and alcohol and tobacco use (5). Thus, poor levels of HL are associated with insufficient use of health services, higher morbidity and hospitalization rates, worse understanding of treatment and health information, and medication errors and have a greater effect on patients with chronic diseases and older adults [13].

The Health Literacy Survey-European Union (HLS-EU) study [14] showed that 35.2% of people consulted had a problematic level of HL reaching 50.8% in Spain. This is an aspect that should be taken into account, especially in patients with complex health problems, such as cardiovascular disease, which require demanding self-care through the modification of cardiovascular risk factors and often complex pharmacological treatment [15,16,17].

To mitigate health inequalities related to HL, it is necessary to know the distribution of HL along the axes of inequality. With that perspective, the aim of this study was to explore the relationships between the dimensions that make up health literacy and the social determinants of health in a sample of patients with cardiovascular disease.

## 2. Methods

### 2.1. Studio Design

Observational, descriptive, cross-sectional study.

### 2.2. Population

We focused on 6 areas within the Xàtiva/Ontinyent Health Department (Valencia, Spain) selected at random from the 17 areas that make up the department. All patients attending booked or walk-in nursing clinics during the study period (1 January 2018 to 30 April 2019) were asked to complete an HL screening questionnaire. Recruitment concluded when the minimum necessary study sample was reached. For a population N = 730, we calculated a minimum representative sample of 252 patients (95% CI and 5% error).

Patients eligible to participate in the study were those aged 50–85 years with cardiovascular pathologies, mainly arrhythmias or valvulopathies, and with an established pharmacological treatment. Exclusion criteria included visual or hearing impairments that prevented completion of the questionnaire, illiteracy, and serious neurocognitive or mental health problems preventing the patient from understanding their pathology.

The study was carried out in rural areas with populations between 2300 and 8000 inhabitants per primary care centre, and where the distance to the health centre was usually close and allowed access on foot for most people who could walk.

### 2.3. Study Assessment Parameters

Data were obtained on the following variables:

Socio-demographic: Self-perceived social class (low/medium/high), gender, age, and educational level (without studies/basic education/university education).

Clinical: Main diagnosis, obesity (BMI > 30) and polymedication (prescription ≥ 5 drugs).

HL: The Health Literacy Questionnaire HLQ [18], which has been already translated and validated for Spanish-speaking persons, was used to assess the level of HL. A 9-factor CFA model was fitted: *p* < 0.000, Comparative Fit Index (CFI) = 0.936, Tucker Lewis Index (TLI) = 0.930, Root Mean Square Error of Approximation (RMSEA) = 0.076, and Weighted Root Mean Square Residual (WRMR) = 1.698. The questionnaire includes 44 items and assesses 9 different dimensions:oDimension 1 (D1): Feeling understood and supported by health care providers.oDimension 2 (D2): Having enough information to manage my health.oDimension 3 (D3): Actively managing my health.oDimension 4 (D4): Social health support.oDimension 5 (D5): Assessment of health information.oDimension 6 (D6): Ability to actively participate with health care providers.oDimension 7 (D7): Navigation through the health system.oDimension 8 (D8): Ability to find good health information.oDimension 9 (D9): Understanding health information well enough to know what to do.

The scores for dimensions 1 to 5 are set to 4 values (Strongly Disagree/ Disagree/ Agree/ Strongly Agree) while dimensions 6 to 9 are set to 5 values (Can’t do it or always have difficulty/ sometimes have difficulty/ usually have ease/ always have ease). Each of the nine scales has been found to be highly reliable (composite reliability range from 0.8 to 0.9 for each of the 4- to 5-item scales). The scores were calculated by adding up the score for each item and dividing it by the number of items included in each dimension. Independent scores were established for each dimension, as required by the authors [19,20,21].

### 2.4. Statistical Analysis

Central tendency and dispersion measures were used to analyse quantitative variables. For qualitative variables, absolute and relative frequencies, expressed as percentages, were used.

Parametric (Anova, post-hoc, and T-Student tests) and non-parametric (Mann–Whitney U and Kruskal–Wallis) tests were carried out to assess the relationship between the different socio-demographic variables and the different dimensions of HLQ. The level of statistical significance was established at *p* < 0.05, and for certain variables, the size of the effect was analysed using Cohen’s d. All data were analysed with Statistical Package for the Social Sciences(SPSS) version 23, Spanish. 

### 2.5. Ethical Considerations

The study was approved by the Ethics and Clinical Research Committee of the Primary Care Region of Valencia (Ref.ACC-ACE-2016-01). Written informed consent was obtained from the participants. Data collection was performed by the principal investigator.

## 3. Results

### 3.1. Description of the Sample

The response rate was 35% (252 patients responded to the questionnaire). The main diagnosis was atrial fibrillation (75%), followed by valvular prosthesis, which represented 23%. The patients were mostly male (58%) and had an average age of about 75 years. About ninety per cent were older than 65 years (Table 1)

### 3.2. Relationships between HL Dimensions and Variables

Table 2 shows the average scores for each HL dimension. In all of them, patients obtained average scores slightly higher than the average score determined by the questionnaire but mainly in D4 (social support). The lowest average score was obtained in D5 (assessment of health information).

In D1 “Feeling understood and supported by health care providers”, D4 “Social support for health”, and D6 “Ability to actively participate with health care providers”, the statistically significant variables were social class and educational level (Table 3).

For D1, social class explained 2% of the variance in scores, constituting a small effect; post-hoc tests showed statistically significant differences (*p* < 0.05) in scores between lower- and middle-class people. In D4, post-hoc tests only reported statistically significant differences (*p* < 0.05) between lower- and upper-class people. In contrast, in D6, they reported statistically significant differences (*p* < 0.05) between the lower- and middle-class group, as well as the lower- and high-class group. The scores of the lower-class participants were lower than those of the middle- or upper-class participants.

With respect to the level of education, it explained 7% of the variance in scores, constituting a small effect on D1. In this case, the post-hoc tests showed statistically significant differences (*p* < 0.05) between those with no studies and those with basic studies, as well as between persons without studies and those with university/higher studies. However, the differences between the group with basic education and the group with university/higher education were not statistically significant (*p* > 0.05).

In D4, the level of education explained 3% of the variance in scores, constituting a small effect. The post-hoc tests showed statistically significant differences (*p* < 0.05) in D4 scores between those with no education and those with university/higher education, while the differences in scores between the group with no education and the group with basic education were statistically nonsignificant (*p* > 0.05). This was also the case for the differences between the group with basic studies and the group with university/higher studies. In contrast, D6 shows statistically significant differences (*p* < 0.05) between the three education level groups.

In the dimensions D2 “To have enough information to manage my health” and D3 “To actively manage my health”, the variables educational level, social class, and polymedication were statistically significant (Table 3).

Social class in D2 explains 8% and in D3, 9% of the variance in scores, representing an average effect in both dimensions. Post-hoc tests report statistically significant differences (*p* < 0.05) between the lower- and middle-class group, as well as between the lower-class and high-class group. The median scores on the respective dimensions are lower in the lower-class group than in the middle-class and upper-class groups.

With regard to the level of studies, the post-hoc tests showed statistically significant differences (*p* < 0.05) between the three study level groups, with the group without studies obtaining the lowest average scores on these dimensions and the university/higher education group obtaining the highest average scores.

For polymedication, t-tests show that there are statistically significant differences (*p* <.0 05) in the scores of both dimensions depending on whether polymedication exists or not. The average scores of patients who do not take five or more drugs are higher than those who are polymedicated. Additionally, the effect size calculated by Cohen’s d shows small effects of polymedication on the scores of these dimensions.

For D5 “Assessment of health information”, the variables age, level of education, social class, polymedication, and obesity had a statistically significant relationship.

Social class explained 13% of the variance in scores, representing a large effect. Statistically significant differences (*p* < 0.05) appeared between all social class groups. The median scores of the lower-class participants were lower than those of the middle-class participants, and, in turn, the median scores in the middle class were lower than those obtained by participants from the higher socioeconomic stratum.

Regarding the level of studies, as in D2 and D3, significant differences were shown between the three study level groups, with the group without studies obtaining the lowest average scores.

Obese participants presented statistically significant differences (*p* < 0.05), showing lower mean scores than the rest of the participants, and the polymedication variable behaved the same as in D2 and D3. In terms of age, higher age is related to lower scores in that dimension.

In D7 “Navigation through the health system”, the same results are obtained as those discussed in D5, with the exception that in this dimension, the obesity variable did not present statistically significant differences (*p* > 0.05).

The dimensions D8 “Ability to find good health information” and D9 “Understanding health information sufficiently to know what to do” were the only ones that showed statistically significant differences between all the studied variables (age, sex, education level, social class, polymedication, and obesity).

In D9, the social class showed significant differences between all social class groups, as in D5. In contrast, D8 only reported differences between the lower- and middle-class group, as well as between the lower- and upper-class group.

Regarding the level of studies, there were statistically significant differences between the three study level groups according to the results of the post-hoc tests. The “without studies” group had lower mean scores on all dimensions than the “basic studies” group, and the “basic studies” group had lower mean scores on all dimensions than the “higher education“ group.

With respect to sex, the results of t-tests show (Table 3) statistically significant differences in the average scores of D8 and D9. On average, men score higher than women on these dimensions. This difference represents a small effect for both D8 (*d* = 0.27) and D9 (*d* = 0.32).

The variables age, polymedication, and obesity behaved in both dimensions exactly the same as in dimensions D2, D5 and D7.

## 4. Discussion

In our study of patients with cardiovascular pathology, we observed that certain sociodemographic variables influenced several dimensions of HL. These results are similar to existing studies linking sociodemographic characteristics of populations, reduced levels of HL, and increased risk of health disparities [8,22,23].

In our study, we assessed the social determinants that make up the axes of inequality and their involvement in HL to clarify whether measuring HL would facilitate addressing health inequalities and increase health equity. Social class was significant to all dimensions of HL, and participants from less privileged social classes obtained lower scores and less capacity to use preventive services in comparison to those from the upper class. These results agree with findings of various studies and refer to the fact that low social class contributes to lower economic income as it influences employment possibilities, providing an inverse relationship between socio-economic position and morbidity and mortality outcomes [24,25,26,27].

Various studies claim that cardiovascular disease outcomes differ by gender [28,29,30]. The 2011 USA report reflected how mortality from heart attack was higher in young women than in men, although recently, such mortality has decreased three times more among women than men, largely because of knowledge and control of risk factors in women [29]. These results resemble those in our study, where men are better able to find information from different sources and are able to understand written health information better than women. O’Neil et al. [31] report that not only does gender imply biological sex but it also interacts with other social determinants such as socioeconomic status or ethnicity and influences cardiovascular health by modulating behaviour with respect to cardiovascular risk factors. Gender is a well known, key structural determinant of health, and our results support the need to stimulate policies to reduce this type of inequality.

Concerning age, recent studies in heart failure patients inversely relate age to HL scores [12,32,33,34]. In our study, equally, older patients were less able to navigate the health system to address their needs on their own, did not understand most health information, had great difficulty filling out medical forms and were confused before conflicting information, depending on others to enable the access to health resources. This scenario is problematic and highlights a cascade of inequality where one in five persons in Spain—almost 9 million people—is over 65 years old. And of them, ~60% are women; and of these, almost 2 million live in solitude. In Europe, the situation could be more concerning, since the number of people in single-person households is almost double that of Spain (7.1% in Spain compared to 13.4% in Europe) [35].

Therefore, based on our results and anticipating a likely worsening of the current situation (by 2068, older adults and the elderly will account for 29.4% of the Spanish population), in the short term, we advocate for the promotion and strengthening of community support networks for older adults, or the development of other neighbourhood-based, community models of coexistence (cohousing, collaborative residential models, supervised housing...). These proposals could reduce the risks for older adults, especially those living in or at greatest risk of loneliness and isolation); in the long term, however, the most useful intervention will be to develop strategies to make health care systems more health literate so that people have the same opportunities for access and use regardless of their level of HL and are not disadvantaged, thus seeking to avoid or reduce inequalities [36].

The environment in which the study was conducted was mostly rural. Participants obtained average scores that were higher than expected, above all in the dimension that values social support. These results coincide with various studies carried out in Spain in populations older than 65 years old that underpin the social environment as a common determinant and that the social networks of family and neighbourhood support tend to be more present or stronger in the rural environment. It seems logical, then, to propose that the development of the community and its assets should be prioritised for public and community health services, based on the strengthening of social support networks, both formal and informal. This is a remarkable factor that will contribute to improving the social support dimension and HL in general, or reducing the consequences of low HL. It remains to be clarified whether the situation is similar in urban settings, or whether these show lower HL scores and, specifically, in the dimension that reflects social support. The existing literature, incidentally, suggests that urban settings inhibit the presence of social micronetworks [37,38,39].

Determinants such as “educational or academic level” have also been directly related to HL level in several studies in heart failure patients conducted between 2014 and 2019 [33,40,41], indicating that academic achievements could explain about 60% of HL [33]. These results coincide with the data obtained in our study, where the level of education was significant for all dimensions of HL. It is a well-known fact that educational level has a powerful influence on social class, mediated by the best jobs that people with high academic levels can get [24,25,26,27]. Both educational level and social class are intermediate determinants studied in our research that have been shown to be related to the level of HL, so raising the educational level of the population must be a political priority. It is well known that the relationship between these determinants (educational level, employment, economic income) contributes to fostering the circle of poverty, a trend that could be broken with the educational improvement that we propose.

HL, in addition to being related to the determinants described above, may be associated with hyperfrequentation of health services and a greater rate of adverse events. This fact is described by different studies in patients with cardiovascular pathology, which associate lower HL scores with higher rates of rehospitalization [42] and increased risk of mortality [42,43,44]. For this reason, we are currently studying this possibility in our study population, since we consider it necessary to know the relationship between HL and certain therapeutic and clinical indicators in patients with cardiovascular pathology.

The dimensions 8 “Ability to find good health information” and 9 “Understanding health information well enough to know what to do’’ are statistically significant for all studied variables. Furthermore, it has been shown that different health determinants such as age over 75 years, basic educational level or no studies, less privileged social classes, women, and multipathological, polymedicated, and obese patients present lower HL scores. Population groups are more vulnerable with greater risk of exposure to the axes of inequality and therefore have worse health outcomes.

Therefore, it is possible to relate HL to the axes of inequality, such as age, gender, studies, and social class, and these elements should also be assessed in people at special risk for low HL, such as people with chronic health problems, often with comorbidity, and polymedication.

## 5. Limitations

In our study, we have established the relationships between the sex variable and the different dimensions of HL, but we have not assessed whether these differences between men and women are amplified by associating social class and educational level as well, an aspect that could have been interesting to evaluate, taking into account that inequalities in health can be cumulative.

In relation to the above, another limitation of the study would be not having been able to carry out a logistic regression analysis because obtaining the average scores of each dimension generates very small population subgroups.

## 6. Conclusions

Our study of patients with cardiovascular pathology has established a strong relationship between the dimensions of HL and the social determinants that make up the axes of inequality in health. It highlights the importance of considering HL as a determinant of health because it is a predictor of individual health status, suggests a social gradient, and through its measurement, facilitates the identification and approach of other health determinants.

Strengthening HL will help identify people in vulnerable situations and address health inequities more effectively. In addition, the supportive interventions implemented need to recognise the social gradient of LH; otherwise, these interventions may not favour those with worse LH and perpetuate health inequities.

## Figures and Tables

**Figure 1 ijerph-17-02082-f001:**
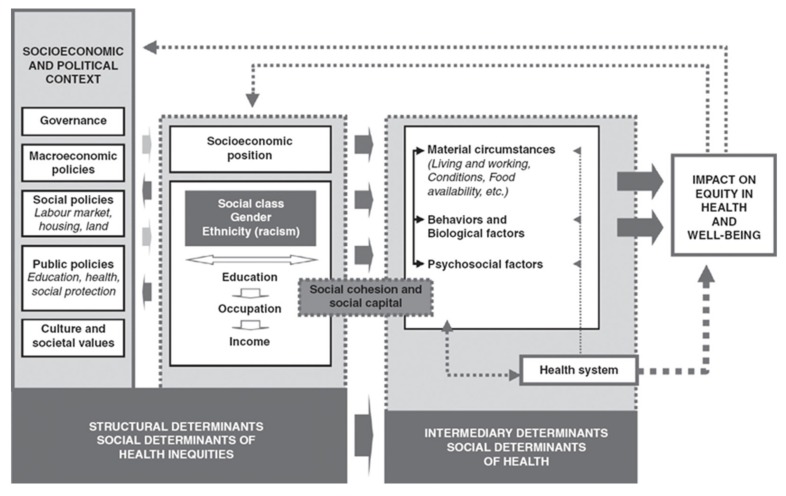
Social determinants of health. WHO Commission on Social Determinants. Source: Solar and Irwin *(3)*.

**Table 1 ijerph-17-02082-t001:** Characteristics of participants.

		M/n	Sd/%
**Age**		74.38	7.35
	< 65 years	26	10.3%
	= > 65 years	226	89.7%
Gender	Woman	107	42.5%
	Man	145	57.5%
Education Level	Without Studies	101	40.1%
	Basic Education	126	50.0%
	Higher Education	25	9.9%
Social Class	Low	33	13.1%
	Middle	206	81.7%
	High	13	5.2%
Main Diagnosis	Atrial Fibrillation	188	74.9%
	Atrial Flutter	7	2.8%
	Aortic Prosthesis	33	13.1%
	Mitral Prosthesis	23	9.2%
Polypharmacy	No	86	34.1%
	Yes	166	65.9%

M = Mean; n = number of cases; Sd = Standard deviation; % = percentage.

**Table 2 ijerph-17-02082-t002:** Average score of HL dimensions.

Dimension	Mean	Standard Error of the Mean	Standard Deviation	Median	Maximum	Minimum	25 Percentile	75 Percentile
Dimension 1	3.201	0.027	0.429	3.125	4.000	1.500	3.000	3.500
Dimension 2	2.657	0.033	0.517	2.750	4.000	1.000	2.250	3.000
Dimension 3	2.817	0.032	0.511	2.800	4.000	1.000	2.600	3.000
Dimension 4	3.494	0.028	0.437	3.600	4.000	1.400	3.200	3.800
Dimension 5	2.387	0.043	0.686	2.400	4.000	1.000	2.000	2.800
Dimension 6	4.106	0.038	0.596	4.200	5.000	1.200	3.800	4.600
Dimension 7	3.507	0.041	0.654	3.667	5.000	1.167	3.083	4.000
Dimension 8	3.052	0.054	0.851	3.200	5.000	1.000	2.400	3.800
Dimension 9	3.268	0.055	0.871	3.400	5.000	1.400	2.600	4.000

**Table 3 ijerph-17-02082-t003:** Relationship and statistical significance between variables and HL dimensions.

		D1	D2	D3	D4	D5	D6	D7	D8	D9
**Social Class ^C^**	***Sig.***	**0.019 ***	**<0.001 ****	**< 0.001 ****	**0.003 ***	**< 0.001 ****	**<0.001 ****	**<0.001 ****	**<0.001 ****	**<0.001 ****
Low	*Mean (SD)*	2.98 (0.44)	2.3 (0.39)	2.41 (0.53)	3.38 (0.39)	1.81 (0.55)	3.76 (0.56)	3.04 (0.58)	2.25 (0.64)	2.43 (0.61)
	*Median (IQR)*	3.00 (0.50)	2.25 (0.50)	2.4 (0.80)	3.40 (0.40)	1.80 (0.80)	3.80 (0.80)	3.00 (0.67)	2.20 (0.40)	2.20 (0.80)
Middle	*Mean (SD)*	3.23 (0.4	2.71 (0.51)	2.87 (0.49)	3.5 (0.45)	2.45 (0.66)	4.15 (0.59)	3.56 (0.65)	3.14 (0.82)	3.35 (0.83)
	*Median (IQR)*	3.25 (0.50)	2.75 (0.75)	3.00 (0.60)	3.60 (0.60)	2.60 (0.80)	4.20 (0.80)	3.67 (0.83)	3.20 (1.20)	3.40 (1.40)
High	*Mean (SD)*	3.27 (0.33)	2.77 (0.5)	3 (0.34)	3.72 (0.29)	2.88 (0.67)	4.34 (0.51)	3.91 (0.31)	3.66 (0.64)	4.10 (0.56)
	*Median (IQR)*	3.25 (0.50)	2.75 (0.25)	3.00 (0.20)	3.80 (0.0)	3.00 (0.40)	4.20 (0.60)	4.00 (0.17)	4.00 (0.80)	4.20 (0.20)
**Gender ^A^**	*Sig.*	0.496	0.055	0.298	0.137	0.096	0.217	0.142	**0.028** *	**0.015** *
Woman	*Mean (SD)*	3.18 (0.46)	2.58 (0.54)	2.78 (0.53)	3.44 (0.5)	2.3 (0.7)	4.05 (0.64)	3.44 (0.66)	2.92 (0.84)	3.11 (0.85)
	*Median (IQR)*	3.00 (0.50)	2.50 (0.75)	2.80 (0.40)	3.60 (0.60)	2.40 (1.00)	4.00 (1.00)	3.50 (1.00)	3.00 (1.40)	3.20 (1.40)
Man	*Mean (SD)*	3.22 (0.4)	2.71 (0.5)	2.85 (0.5)	3.53 (0.38)	2.45 (0.67)	4.15 (0.56)	3.56 (0.65)	3.15 (0.85)	3.38 (0.87)
	*Median (IQR)*	3.25 (0.50)	2.75 (0.75)	2.80 (0.60)	3.60 (0.40)	2.40 (0.80)	4.20 (0.80)	3.67 (0.83)	3.20 (1.20)	3.40 (1.40)
**Age ^D^**	*Sig.*	0.022	−0.105	−0.104	−0.030	−**0.223** **	−0.118	−**0.189** *	−**0.282** **	−**0.261** **
<65 years	*Mean (SD)*	3.18 (0.4)	2.73 (0.47)	2.86 (0.58)	3.48 (0.33)	2.59 (0.78)	4.35 (0.49)	3.74 (0.58)	3.42 (0.76)	3.59 (0.85)
	*Median (IQR)*	3.13 (0.50)	2.75 (0.50)	3.00 (0.80)	3.40 (0.60)	2.60 (1.20)	4.40 (1.00)	3.83 (0.67)	3.50 (1.20)	3.80 (1.60)
=>65 years	*Mean (SD)*	3.2 (0.43)	2.65 (0.52)	2.81 (0.5)	3.5 (0.45)	2.36 (0.67)	4.08 (0.6)	3.48 (0.66)	3.01 (0.85)	3.23 (0.87)
	*Median (IQR)*	3.13 (0.50)	2.75 (0.75)	2.80 (0.40)	3.60 (0.60)	2.40 (0.80)	4.20 (0.60)	3.50 (1.00)	3.00 (1.20)	3.40 (1.60)
**Education Level ^B, C^**	***Sig.***	**<0.001 ****	**<0.001 ****	**<0.001 ****	**0.016 ***	**<0.001 ****	**<0.001 ****	**<0.001 ****	**<0.001 ****	**<0.001 ****
Without Studies	*Mean (SD)*	3.06 (0.4)	2.33 (0.43)	2.51 (0.45)	3.42 (0.41)	1.94 (0.54)	3.85 (0.60)	3.08 (0.54)	2.38 (0.57)	2.56 (0.58)
	*Median (IQR)*	3.00 (0.25)	2.25 (0.5)	2.60 (0.60)	3.60 (0.60)	2.00 (0.80)	4.00 (0.60)	3.17 (0.83)	2.40 (0.80)	2.40 (0.80)
Basic Education	*Mean (SD)*	3.28 (0.44)	2.83 (0.47)	2.96 (0.42)	3.52 (0.47)	2.58 (0.56)	4.23 (0.53)	3.72 (0.57)	3.39 (0.66)	3.62 (0.67)
	*Median (IQR)*	3.25 (0.50)	3.00 (0.5)	3.00 (0.40)	3.80 (0.40)	2.60 (0.60)	4.20 (0.60)	3.83 (0.50)	3.60 (1.00)	3.80 (0.60)
Higher Education	*Mean (SD)*	3.4 (0.36)	3.11 (0.32)	3.36 (0.39)	3.67 (0.26)	3.26 (0.56)	4.54 (0.49)	4.16 (0.42)	4.06 (0.63)	4.35 (0.49)
	*Median (IQR)*	3.25 (0.50)	3.00 (0.25)	3.20 (0.80)	3.80 (0.20)	3.20 (1.00)	4.60 (0.60)	4.00 (0.33)	4.00 (0.80)	4.40 (0.60)
**Obesity ^A^**	*Sig.*	0.895	0.288	0.113	0.525	**0.030 ***	0.318	0.142	**0.030 ***	**0.018 ***
No	*Mean (SD)*	3.2 (0.43)	2.69 (0.52)	2.86 (0.51)	3.48 (0.45)	2.46 (0.7)	4.14 (0.55)	3.56 (0.65)	3.15 (0.84)	3.38 (0.86)
	*Median (IQR)*	3.13 (0.50)	2.75 (0.75)	2.80 (0.60)	3.60 (0.60)	2.60 (1.00)	4.20 (0.80)	3.67 (0.83)	3.20 (1.30)	3.40 (1.30)
Yes	*Mean (SD)*	3.2 (0.42)	2.62 (0.52)	2.76 (0.51)	3.52 (0.41)	2.28 (0.65)	4.06 (0.66)	3.43 (0.66)	2.91 (0.86)	3.11 (0.86)
	*Median (IQR)*	3.13 (0.50)	2.50 (0.75)	2.80 (0.50)	3.60 (0.50)	2.40 (1.00)	4.20 (0.60)	3.50 (0.83)	3.00 (1.50)	3.20 (1.50)
**Polypharmacy ^A^**	*Sig.*	0.255	**<0.001 ****	**0.005 ***	0.780	**<0.001 ****	0.367	**0.004 ***	<**0.001 ****	<**0.001 ****
No	*Mean (SD)*	3.24 (0.4)	2.82 (0.46)	2.94 (0.58)	3.48 (0.4)	2.62 (0.68)	4.15 (0.63)	3.67 (0.64)	3.4 (0.83)	3.58 (0.84)
	*Median (IQR)*	3.25 (0.50)	3.00 (0.50)	3.00 (0.40)	3.60 (0.60)	2.60 (0.60)	4.20 (0.80)	3.83 (0.50)	3.60 (1.20)	3.80 (1.00)
Yes	*Mean (SD)*	3.18 (0.44)	2.57 (0.53)	2.75 (0.46)	3.5 (0.46)	2.26 (0.66)	4.08 (0.58)	3.42 (0.65)	2.87 (0.81)	3.10 (0.84)
	*Median (IQR)*	3.00 (0.50)	2.50 (0.75)	2.80 (0.40)	3.60 (0.60)	2.40 (1.00)	4.10 (0.60)	3.33 (0.83)	2.80 (1.40)	3.10 (1.40)

* *p* < 0.05; ** *p* < 0.001; A = T-Student tests; B = Anova; C = Kruskal–Wallis; D = Mann–Whitney U; SD = Standar desviation; IQR = Interquartile range.

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
