# Peer review of "Relationship between Determinants of Health, Equity, and Dimensions of Health Literacy in Patients with Cardiovascular Disease"

_ijerph, 2020, doi:10.3390/ijerph17062082_

Round 1
Reviewer 1 Report
Thank you for possibility to review this paper. Study cover important health related issue.
Some my questions, comments.
It is just comment. Focusing on health literacy it is very popular to quote HLS-EU study. However, I am very critical of it because of the very questionable sample sizes and sampling technique. But, I understand, that we have to cite this study if we do not have something new (regarding some particular country data).
Population. What do you mean by saying "were selected at random" (lines 105-106). I think just to say random is not enough. Did you invited all people attending nursing consultations to participate in this survey? Final sample was 252 as you needed by calculating sample size (strange if you invited all people participate in the survey, or I did not clearly understand this). But response rate was 35%, it is very low. How can you explain this.
The Health Literacy Questionnaire was used in this study (lines 120-121). I am not sure, was it validated in Spanish? And why did you mention just CFI value from previous study (Osborne et al., 2013)?
All statistical analysis just comparison (t-test, ANOVA, Mann-Whitney). Wonder why was not used other statistical methods, as now you can not talk about predictor in conclusion.
Discussion section just comparison with previous studies. I wonder it is not enough just to make comparison. The main my concern, I missed some practical recommendation. Or, it is difficult to recommend something based on these results?
Author Response
We submit a word archive. Thanks!

Reviewer 2 Report
The topic of the draft, health literacy (HL) research in cardiovascular patients, is always timely and should be kept on the agenda. The results of this research are as expected: patients with low education and social status have lower HL according to the survey. In the introduction, I perceive some conceptual confusions as it seems that the authors value HL over other determinants, although HL is itself depends on social and income status, as the cited literature emphasizes (Solid facts, WHO, 2013). I believe that the manuscript in its present form is not innovative enough and could be supplemented in several directions. The paper covers elderly people in a rural environment with traditional family networks; it would be useful to check urban population as well in a comparative study. Another option is to carry out a longitudinal, prospective study, in which some groups receive detailed health information on CVD prognosis, relapse prevention, while others receive only the usual minimum and clarify how mortality develops in these groups. Of particular note is access to care according to the level of health literacy.
Author Response
We submit a word archive. Thanks!

Reviewer 3 Report
I think this is an excellent and unusual study that will be of interest to diverse readers in stimulate more research on the role of HL in social inequities in health and healthcare.
I have two inter-related conceptual concerns, one comment on the methods sections and some thoughts on improving the analysis and presentation.
First, with respect to the model you adopt, I believe social (race/ethnicity, gender, class etc.) inequalities in health are created through three sets of factors: individual knowledge attitudes and behavior; exposure to unequal physical and built environments; and exposure to unequal health care. This is the standard understanding in the literature of the concept of social determinants of health....your use that equates social determinants with individual membership in unequally treated/served social groups but does not include behavior by health system actors and physical/built environments is unusual and should be more clearly acknowledged.
Closely connected to this concern is my worry about the final outcome in the figure 1 model. HL describes an important set of client experiences, expectations, and self-ascribed capabilities but this is not the same as the differences that population groups may experience. None of this is disconnected from where people live....I think your study would be strengthened to whatever extent you can talk about place....either from the perspective of considering additional covariates you may also have available like residential types....or including measures of variation in distance from the medical providers or walkability of communities....or at least describing your place in more detail so that readers could understand the context and significance of relatively high levels of HL.
Methodologically: You do not provide adequate information about the covariates.....I do not know what the levels of social class refer to and how they were discerned. I am unclear why you only show an above and below 65 dichotomy in your descriptive table....there may for example be a significant group in their 70's and 80's here and their high level of HL would be an important finding...or differences in HL by age could be assessed more fully. The obesity variable is unclear....Are the dichotomies in Table 1 the way the variables were coded for the analyses in Table 3?
My primary concern about the analysis of effects of covariates (age, gender, education, class etc) and the long section in results on page 8/12 is that it is too much detail. and does not seem justified by the initial results...One suggestion: could you form a summary measure of HL combining the dimensions and then test its relationships to the covariates, present this finding in the text, direct attention to an appendix with the details of current table 3.....the text could also highlight if there are any unusual deviations from the overall result.
Author Response
We submit a word archive. Thanks!

Round 2
Reviewer 2 Report
The draft has improved a lot, especially the discussion section, which carefully outlines the meaning, limitations and usefulness of the research. As stated earlier, it would be worth exploring the relationship between HL and access to care, as this is and will remain a major challenge especially for single elderly people.
Author Response
Enrique Castro-Sánchez
NIHR Health Protection Research Unit
in Healthcare Associated Infection and Antimicrobial Resistance
at Imperial College London,
London W12 0NN
6th March 2020
Editorial Office
International Journal of Environmental Research and Public Health
Manuscript “Relationship between determinants of health, equity, and dimensions of health literacy in patients with cardiovascular disease” by Cabellos-García et al.
We have the pleasure in submitting a revised version of the above-mentioned manuscript following the report of the reviewer. We are grateful for their input and comments, which have strengthened the paper.
In response to your comments, in future research we will continue to explore the relationship between health literacy and access to health care more thoroughly, as it could be of great interest to older people living alone.
We would like to thank you in advance for your consideration of our work and we look forward to the comments of the editorial committee in due course.
Best wishes,
Dr Enrique Castro-Sánchez PhD MPH BSc RGN DipTropNurs PgDip DLSHTM FEANS
Lead Academic Research Nurse
NIHR Health Protection Research Unit in Healthcare Associated Infection and
Antimicrobial Resistance at Imperial College London
Honorary Consultant Nurse in Communication & Patient Engagement
Imperial College Healthcare NHS Trust
|
Comments reviewer 2 |
Response |
|
The draft has improved a lot, especially the discussion section, which carefully outlines the meaning, limitations and usefulness of the research. As stated earlier, it would be worth exploring the relationship between HL and access to care, as this is and will remain a major challenge especially for single elderly people. |
We are very grateful for your comment, we have worked to improve the draft and we are very grateful for the input you have given us. The project is planned to be carried out in other areas of Valencia, some of them more urban, so we will take into account the aspect of access to health services in a more comprehensive way.
In Spain the health system is usually very accessible to the entire population because there are figures such as the "case management nurse" or the "social worker" who are responsible for identifying potential patients at risk of loneliness, more complex or with difficulties in accessing health systems. For these patients, specific interventions are carried out and care is provided in their own homes so that they do not have to travel to healthcare centres. In this way, inequalities in access are reduced.
However, in future research we will assess the ease of access to healthcare centres, distance and the presence of informal carers... together with health literacy to explore this aspect in depth.
|
Reviewer 3 Report
I think the paper has been much improved.
I am still troubled by the theoretical figure and associated discussion. 1) the outcome variable is inequalities in health but HL does not appear. 2) HL is a product of social location factors and health system factors---treating it as a preferred measure of social inequalities discounts the potential role of the health and human services in building health literacy...the presentation does not address these concerns.
I am still troubled by the analysis and presentation.....there are 9 dependent variables and 6 independent variables so 54 tests, meaning that a much higher level for significance is needed....The authors should either consider a summary variable and regression analyses, regression analyses on each dv, or at least a Bonferroni or similar correction....At the very least the authors should note in their discussion the impact of multiple tests and offer some argument as to why many of the seemingly significant results should be interpreted given this concern.
The paper needs a quick review and edit for English usage....there are a number of sentences with too many clauses that are harder to understand than they should be.
Author Response
Enrique Castro-Sánchez
NIHR Health Protection Research Unit
in Healthcare Associated Infection and Antimicrobial Resistance
at Imperial College London,
London W12 0NN
6th March 2020
Editorial Office
International Journal of Environmental Research and Public Health
Manuscript “Relationship between determinants of health, equity, and dimensions of health literacy in patients with cardiovascular disease” by Cabellos-García et al.
We have the pleasure in submitting a revised version of the above-mentioned manuscript following the report of the reviewer. We are grateful for their input and comments, which have strengthened the paper.
In response to your comments, we have clarified the issues related to the health determinants model. We emphasize the importance of human and healthcare services in building health literacy. In addition, the difficulty of creating a summary variable of the level of literacy in health, which makes it more difficult to interpret the results, has been included in the section on limitations.
We would like to thank you in advance for your consideration of our work and we look forward to the comments of the editorial committee in due course.
Best wishes,
Dr Enrique Castro-Sánchez PhD MPH BSc RGN DipTropNurs PgDip DLSHTM FEANS
Lead Academic Research Nurse
NIHR Health Protection Research Unit in Healthcare Associated Infection and
Antimicrobial Resistance at Imperial College London
Honorary Consultant Nurse in Communication & Patient Engagement
Imperial College Healthcare NHS Trust
|
Comments reviewer 3 |
Response |
|
I am still troubled by the theoretical figure and associated discussion. 1) The outcome variable is inequalities in health but HL does not appear. 2) HL is a product of social location factors and health system factors--treating it as a preferred measure of social inequalities discounts the potential role of the health and human services in building health literacy...the presentation does not address these concerns.
|
We greatly appreciate your comments. We agree with you that health literacy does not appear in this figure. This figure was developed by the WHO to describe the social determinants of health and we only use it to frame our work and describe the relationship between the determinants and health inequalities, so HL does not appear.
With regard to point 2, we have stressed in the introduction the idea that health literacy depends on social location factors and on interaction with the health system factors, so it is vitally important to take these aspects into account. (Lines 70-78)
|
|
I am still troubled by the analysis and presentation.....there are 9 dependent variables and 6 independent variables so 54 tests, meaning that a much higher level for significance is needed....The authors should either consider a summary variable and regression analyses, regression analyses on each dv, or at least a Bonferroni or similar correction....At the very least the authors should note in their discussion the impact of multiple tests and offer some argument as to why many of the seemingly significant results should be interpreted given this concern.
|
We are grateful to the reviewer for mentioning this aspect and agree that the elaboration of a summary variable would facilitate the interpretation of the results, but in this case it is impossible. This questionnaire does not allow this and the creators insist that separate scores must be maintained for each dimension at all times because each values a different aspect of HL. (Lines 140-141)
We are aware that sample size and the fact that we cannot calculate a summary variable makes it difficult to compare significant results, so we have included the size of the effect in the interpretation of some variables.
We appreciate and are very grateful for the suggestion to carry out a logistic regression analysis, but currently with the sample we have it is not possible to do so because obtaining the mean scores of each dimension generates very small subgroups of the population. It is envisaged that this will be carried out in future research with larger sample sizes or with stratified samples.
In addition, we are currently obtaining a greater number of independent variables related to ease of access to the health system and to health outcomes.
We have added all these aspects in the section on study limitations.
|
|
The paper needs a quick review and edit for English usage....there are a number of sentences with too many clauses that are harder to understand than they should be. |
Thanks for this contribution. We have checked the language again and tried to modify the more complex sentences to make them easier to understand. The article has been revised by a professional translator who is a member of the translation team at the Health Research Institute of the Hospital La Fe in Valencia.
|